# Measuring the impact of climate change on potato production in Bangladesh using Bayesian Hierarchical Spatial-temporal modeling

**Md. Sifat Ar Salan**[1], **Md. Moyazzem Hossain**[1]*, **Imran Hossain Sumon**[1], **Md. Mizanur Rahman**[2], **Mohammad Alamgir Kabir**[1], **Ajit Kumar Majumder**[1]

1 Department of Statistics, Jahangirnagar University, Savar, Dhaka, Bangladesh, 2 Department of Statistics, Mawlana Bhashani Science and Technology University, Santosh, Tangail, Bangladesh

* hossainmm@juniv.edu

## Abstract

**Data Availability Statement:** This study is based on the secondary data that is freely available in the websites of the Bangladesh Bureau of Statistics

### Background

Potato is a staple food and a main crop of Bangladesh. Climate plays an important role in different crop production all over the world. Potato production is influenced by climate change, which is occurring at a rapid pace according to time and space.

### Objective

The main objective of this research is to observe the variation in potato production based on the discrepancy of the variability in the spatial and temporal domains. The research is based on secondary data on potato production from different parts of Bangladesh and five major climate variables for the last 17 years ending with 2020.

### Methods

Bayesian Spatial-temporal modelling for linear, analysis of variance (ANOVA), and auto-Regressive models were used to find the best-fitted model compared with the independent Error Bayesian model. The Watanabe-Akaike information criterion (WAIC) and Deviance Information Criterion (DIC) were used as the model choice criteria and the Markov Chain Monte Carlo (MCMC) method was implemented to generate information about the prior and posterior realizations.

### Results

Findings revealed that the ANOVA model under the Spatial-temporal framework was the best model for all model choice and validation criteria. Results depict that there is a significant impact of spatial and temporal variation on potato yield rate. Besides, the windspeed does not show any influence on potato production, however, temperature, humidity, rainfall, and sunshine are important components of potato yield rate in Bangladesh.

(http://bbs.portal.gov.bd/sites/default/files/files/bbs.portal.gov.bd/page/b343a8b4_956b_45ca_872f_4cf9b2f1a6e0/45%20years%20Major%20Crops.pdf) and Bangladesh Agricultural Research Council (https://bbs.portal.gov.bd/site/page/453af260-6aea-4331-b4a5-7b66fe63ba61/-).

**Funding:** The author(s) received no specific funding for this work.

**Competing interests:** The authors have declared that no competing interests exist.

## Conclusion

It is evident that there is a potential impact of climate change on potato production in Bangladesh. Therefore, the authors believed that the findings will be helpful to the policymakers or farmers in developing potato varieties that are resilient to climate change to ensure the United Nations Sustainable Development Goal of zero hunger.

## Introduction

Potato (Solanum tuberosum L.) is the most efficient crop and the third most significant food crop on the planet. Bangladesh, which is ranked seventh in the globe, is now a significant producer of potatoes in the SAARC region [1]. To achieve the Sustainable Development Goals (SDGs)-2, zero hunger as well as to ensure the food demand of our country's constantly expanding population, people have relied heavily on the major cereal crops of rice, wheat, and maize in order to ensure food security. In order to maintain food security in the coming decades, it is necessary to reduce our reliance on grains and potatoes can be a suitable alternative. Researchers highlighted that education and training are beneficial for enhancing potato grower's capacity to absorb and comprehend information about contemporary technologies that may help to produce more potatoes [2]. However, the current situation of global potato production unfolds in the context of climate change, which is projected to impair potato production due to rising temperatures, increased atmospheric $CO_2$, shifting precipitation patterns, and more extreme weather events [3]. It is already established that crop production is highly dependent on climate and a previous study pointed out that climate change can reduce 25.7% of total crop production by 2080 [4]. A previous study highlighted the impacts of climate variables on rice production in Bangladesh [5]. However, all the crops are not equally affected by climate indicators. Plant growth, development, and yield are mainly influenced by temperature, and day degrees are commonly employed to assess this effect [6]. Only temperate climates with lengthy days are suitable for growing potatoes because of their extreme thermos-sensitivity. Besides, potato plants are sensitive to water shortages. It seems expected that the present global warming trend (0.6±0.2˚C), which has been observed since 1900, will continue, and that the average world temperature will rise by 1.4 to 5.8°C between 1990 and 2100 [7]. This rise in temperature might have a significant influence on potato farming across the world [8]. Potato is also frost-sensitive, and significant damage can result when the temperature falls below zero degrees Celsius [9]. Moreover, precipitation and the hydrological cycle have received increasing attention recently [10]. Potato might be severely impacted, putting millions of farmers, particularly small and marginal growers, at risk of losing their livelihood and food security. Overall crop productivity will most likely drop as a result of this sort of climate change, although there will be significant regional variances [11].

According to research by the Universal Ecological Fund (FEU-US), rising temperatures would have a significant impact on the world's food output, with subcontinents on the Indian oceans being the most negatively impacted [12]. However, potato trees can flourish in both cold and warm conditions since they can withstand both drought and barren conditions, while they are less resistant to high temperatures and humidity [13]. The distribution of heat and rainfall throughout the winter months in Bangladesh is synchronized with the developing and extending phases of the potato tuber, making conditions ideal for the growth of potatoes. A previous study pointed out that there is an impact of climatic change on the water use efficiency (WUE) of potatoes in the northwest semiarid area of China [14]. The findings showed

that, as a result of climatic warming and a decrease in rainfall during that time, the WUE of potatoes dramatically rose [14]. The majority of crops grown in northern Europe, including potatoes, are anticipated to experience better-growing conditions as a result of warmer temperatures and higher $CO_2$ levels [15], though there was inevitably unfavorable effects that varied in terms of time and space [16]. In northern China, the potato yield was most susceptible to changes in the diurnal temperature range, followed by radiation, precipitation, and evapotranspiration [17], late potato development was sensitive to the optimum temperature setting in northern Europe [18], and spatial variability impacts potato production [19]. The maximum temperature, comparatively heavy rain, and irrigation all had an effect on the yield of potatoes in China [20].

Due to its location smack dab in the middle of the Himalayas and the Bay of Bengal, Bangladesh, a developing economy, is regarded as one of the planet's most climate-sensitive countries. The increased frequency of natural disasters brought on by global warming has a severe influence on crop productivity, both directly and indirectly. A previous study highlighted that Bangladesh is blessed with extraordinarily fertile soil, temperate temperatures, and ample rainfall that enable the year-round production of a range of crops [21]. Bangladesh is expected to find the optimum setting for boosting potato production to meet the growing food demand given its limited agricultural land. Small variations in climatic conditions may be effectively controlled by adjusting planting dates, spacing, and input management. Also, regional variation can have a significant impact on potato production. Different regions do not experience the same environmental changes. As with the environmental variability, the production of potatoes is also varying regionally. It is anticipated that changes in potato production may be related to both geographic location and climate variables. Therefore, the authors aimed to find out the potential consequences of climate change on potato production in different regions of Bangladesh. A Bayesian Hierarchical Spatial-temporal model has been used to examine the variations among the spatial domains and changes over time. The Bayesian spatio-temporal model is primarily based on the Bayesian framework, which takes into account both the location and time domains. Given the large fluctuation in production based on time and location, this model can be particularly useful for modeling agricultural data [22]. It enhances modeling for the agro-climatic environment because the spatio-temporal model only functions for the specific data frame that contains the time variable as well as the space variable. Therefore, the authors believed that the findings will aid researchers looking into how various climate factors affect various agricultural crops around the world.

## Methods and materials

### Data collection, and study area

The secondary data on potato production was extracted from the different Yearbook of Agricultural Statistics of Bangladesh published by the Bangladesh Bureau of Statistics (BBS) for all 64 districts of Bangladesh over the period 2004–2020 [23]. The area of the potato production is measured in the acre and the total production is in Metric tons. Climate variables of 35 weather stations were collected from the Bangladesh Agricultural Research Council (BARC) for the same time interval [24]. The authors used the district's neighbouring weather station information (if that district doesn't have a weather station) to generate the climatic data for 64 districts.

### Variables

The annual total potato production was the combination of local and hybrid (HYV) production. The yield rates were considered as the response variables of the model. The information

on the harvested areas for all two types of production is also included as an exposure to the response. In this study, the annual average of maximum temperature ($^0$C), rainfall (mm), sunshine (hour), wind speed (m/s), and humidity (%) were considered as the covariates. Moreover, the spatial domain was the districts (2$^{nd}$ administrative level) containing 64 unique district names and the temporal domain was the 17 years for the consecutive districts. Therefore, each variable contains 1088 (64×17) observations (rows).

## Bayesian hierarchical model

The response variable, yield rate $E_{dt}$ for $d$ = 1,2,...,64 $and$ $t$ = 1,2,...,17 was assumed to follow the Gaussian distribution, $E_{dt} \sim N(\mu_{dt}, \sigma_{dt}^2)$. Let, $x_{dt}$ indicates the p-dimensional Spatio-temporal covariates at space–time combination $d$ and $t$ that may influence potato yield rate. In order to create explanatory models based on covariates and structured spatio-temporal random effects is denoted by $\psi_{dt}$ and the model can be expressed as,

$$E_{dt} \sim x_{dt}'\beta + \psi_{dt}, \tag{1}$$

for $d$ = 1,2,...,64 $and$ $t$ = 1,2,...,17. The random effects $\psi_{dt}$ are assumed to have the following diverse range of models based on various assumptions regarding their architecture. To select the appropriate model, Bayesian model selection criteria such as the Deviance Information Criterion (DIC) [25] and Watanabe-Akaike information criterion (WAIC) [26] were used in this study. The general spatial-temporal models were considered as,

$$\psi_{dt} = \begin{cases} \phi_d + \delta_t + \gamma_{dt}, & Anova\ Model \\ \beta_1 + \phi_d + (\beta_2 + \delta_d)\dfrac{t - \bar{t}}{T}, & Linear\ Model\ of\ Trend \\ \phi_{dt} + \delta_t, & Auto\ Regressive\ Model \end{cases}$$

for $d$ = 1,2,...,64 $and$ $t$ = 1,2,...,17. In the ANOVA model, $\phi$, $\delta$ and $\gamma$ are sets of random effects parameters and let $\phi = (\phi_1,...,\phi_d)$ and $\delta = (\delta_1,...,\delta_t)$. We assume that the distributions of the priors of the models follow conditional autoregressive (CAR) with $\rho_S$, $\rho_T$, $\tau_S^2$, $\tau_T^2$ where these are the auto-regression and variance parameters of the spatial and temporal inference [27].

The hierarchical Bayesian spatial-temporal model has been applied to satisfy the objectives starting with the independent error Bayesian generalized linear model (GLM) and few spatial-temporal models by adding the spatial and temporal domains were considered. The MCMC methods have been implemented to estimate the priors for estimating the Posterior estimation. In the **CARBayes** package [28], the unknown variance parameter in the Gaussian model is provided by default an inverse gamma prior distribution with shape parameter = 1 and scale parameter = 0.01. The **bmstdr** package is used to prepare all the models, plots and maps.

## Ethical statement

Not applicable.

## Results

The authors assumed that $Y_{dt}$ denotes the total potato production for $d^{th}$ district and $t^{th}$ year. Here $d$ = 1,2,...,64 as 64 districts and $t$ = 1,2,...,17 for 17 years from 2004 to 2020. The magnitude of $Y_{dt}$ depends on the consecutive harvested area $Z_{dt}$ of a particular district and year. The yield rate is denoted by $E_{dt}$ and is defined as, $E_{dt} = \frac{Y_{dt}}{Z_{dt}}$; $d$ = 1,2,...,64 $and$ $t$ = 1,2,...,17. Similar measures have been used to estimate for local and HYV production $E_{dtL}$ and $E_{dtH}$.

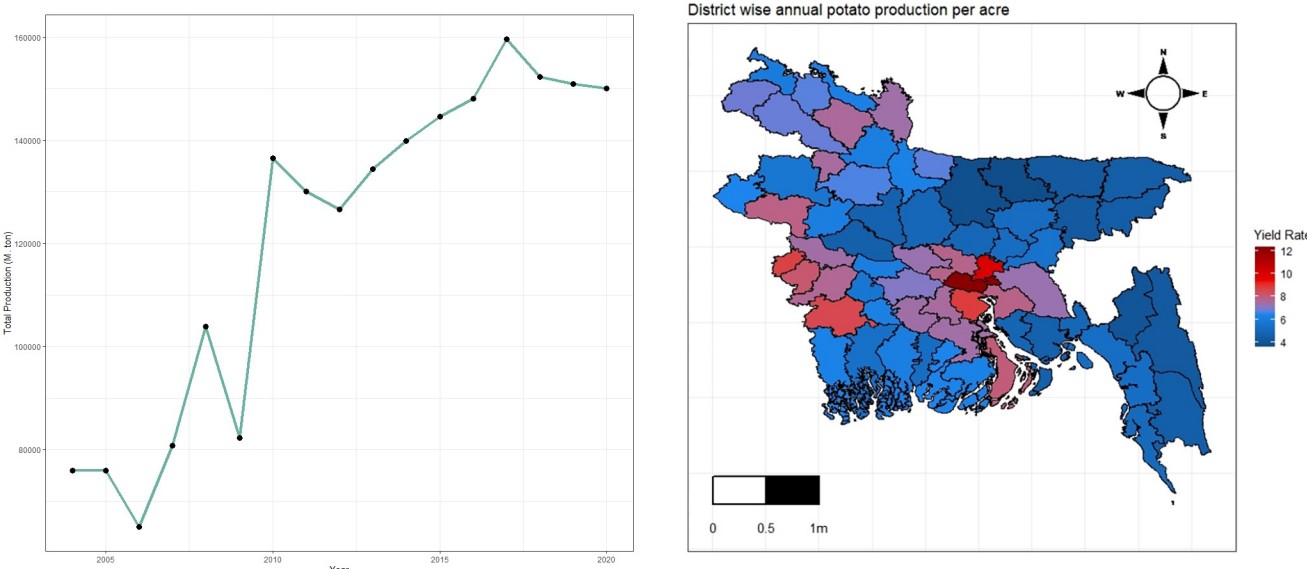

**Fig 1.** Potato production in Bangladesh, (a) time series plot, (b) spatial distribution.

It is observed that the total production of potatoes is increasing over the last two decades in Bangladesh [Fig 1(A)]. The maximum production of potatoes in Bangladesh was 1,59,624.38 metric tons in 2017 and the Munsiganj district was on the top in annual total potato production and yield rate within the time frame. In 2016, the farmers of Munsiganj produced 12,42,329 metric tons of potatoes which was the district level maximum from 2004 to 2020. However, the maximum area was used to harvest the potato from the Bogura district [Fig 1 (B)].

To explore the relationship between potato production and the climate variables, the authors used a pairwise scatter plot presented in Fig 2. The square root of the temperature and the log of the rainfall has been used to adjust the scale with the target variable. The numerical values of the plot revealed the existence of various moderate associations among variables. A significant positive association is observed between temperature and yield rate, however

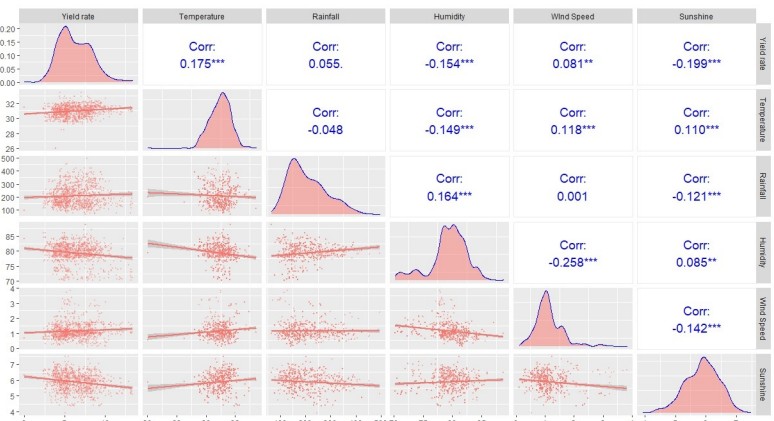

**Fig 2. Matrix scatter plot of the selected variables.** *** refers p-value <0.001, ** refers p-value <0.05 and * refers p-value <0.1.

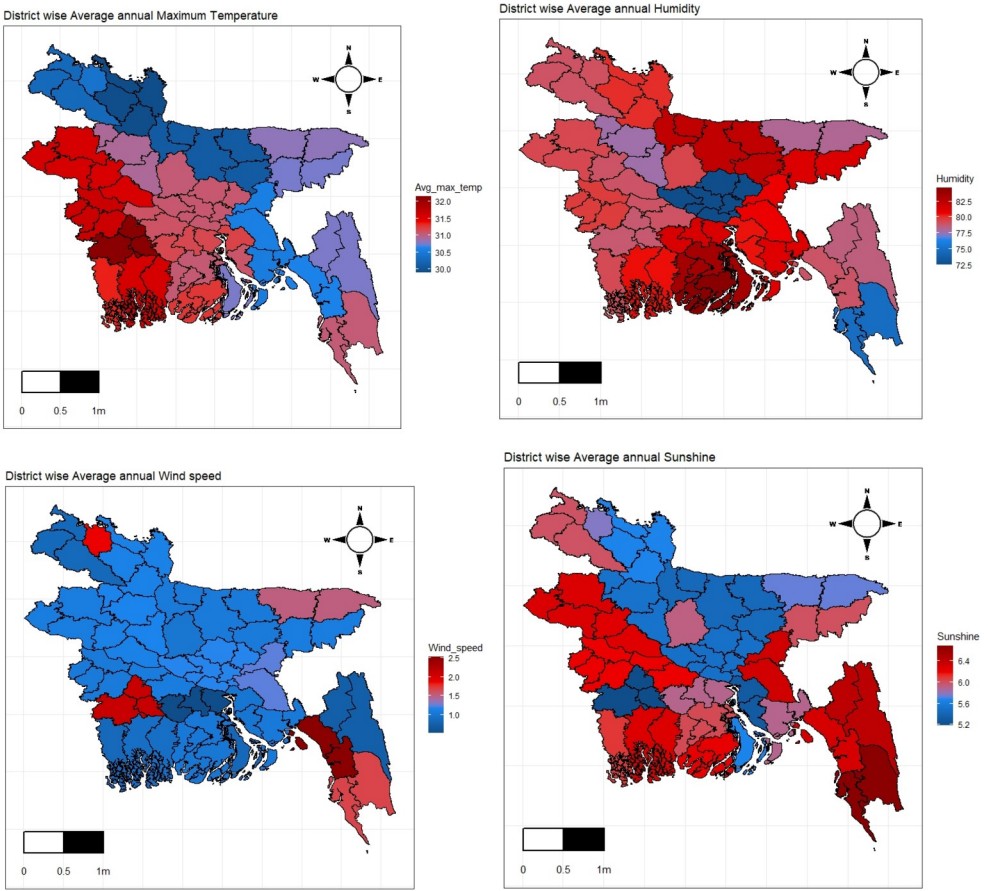

**Fig 3.** District wise variation of climate variables (a) maximum temperature, (b) humidity, (c) wind speed, and (d) sunshine.

humidity and sunshine having a significant negative association with it. The rainfall is not correlated with the yield rate and wind speed has a weak relationship with yield rate [Fig 2].

The nature of the significant covariates according to the spatial domain is also displayed in Fig 3. It is observed that all the figures show a significant variation in-between different regions of the Bangladesh. However, the capital (Dhaka) and Bangladesh's coastal region had relatively low humidity.

The strength of humidity is a bit high in the northeast part compared to other regions and the opposite scenario is visible on the map of temperate. It is also seen that both have a different association with the potato yield rate. The windspeed is almost uniform over the whole country except for the coastal area and the sunshine is as like as the temperature [Fig 3].

Before starting the modelling with the spatial and temporal domain, the authors preferred to begin with an independent error Bayesian model so that it can be understood the improvement when the spatial and temporal effects are incorporated into the model. The linear predictor $x'_{dt}\beta$ can be specified in the model as,

$$temperature + rainfall + humidity + windspeed + sunshine$$

where the temperature is in square root scale and rainfall is in log scale. All the models were run under 100,000 iterations after 20,000 burn-in iterations. In the MCMC context, the data were stored after a thinning of 10 iterations to decrease autocorrelation.

**Table 1. Estimated parameters with their credible interval for all proposed models.**

| | Independent Error Bayesian GLM | Bayesian Spatial-temporal Models | | |
| --- | --- | --- | --- | --- |
| | | Linear Model | ANOVA Model | AR Model |
| | Estimate | Estimate | Estimate | Estimate |
| | (Credible Interval) | (Credible Interval) | (Credible Interval) | (Credible Interval) |
| Intercept | -15.463 | -15.539 | -11.234 | -13.316 |
| | (-26.441, -4.545) | (-26.71, -4.089) | (-22.797, -0.311) | (-24.652, -1.49) |
| Temperature | 5.594 | 5.7 | 4.823 | 5.134 |
| | (3.801, 7.435) | (3.826, 7.607) | (2.867, 6.752) | (3.173, 7.08) |
| Rainfall | 0.372 | 0.46 | 0.829 | 0.824 |
| | (-0.356, 1.108) | (-0.297, 1.221) | (0.016, 1.638) | (0.004, 1.637) |
| Humidity | -0.073 | -0.074 | -0.078 | -0.074 |
| | (-0.111, -0.035) | (-0.114, -0.034) | (-0.121, -0.036) | (-0.116, -0.032) |
| Windspeed | 0.005 | -0.038 | 0.06 | 0.019 |
| | (-0.239, 0.244) | (-0.288, 0.21) | (-0.193, 0.307) | (-0.232, 0.269) |
| Sunshine | -0.763 | -0.859 | -0.859 | -0.858 |
| | (-0.978, -0.545) | (-1.09, -0.625) | (-1.1, -0.63) | (-1.089, -0.621) |
| $\tau_S^2$ | - | - | 0.204 | - |
| | | | (0.105, 0.416) | |
| $\tau_T^2$ | - | - | 0.377 | 0.272 |
| | | | (0.192, 0.822) | (0.134, 0.564) |
| $\nu^2$ | 3.786 | 3.632 | 3.409 | 3.366 |
| | (3.488, 4.128) | (3.331, 3.986) | (3.121, 3.724) | (3.07, 3.704) |
| $\rho_S$ | - | - | 0.444 | 0.982 |
| | | | (0.058, 0.896) | (0.935, 0.994) |
| $\rho_T$ | - | - | 0.256 | 0.094 |
| | | | (0.013, 0.778) | (0.004, 0.389) |
| $\alpha$ | - | -0.823 | - | - |
| | | (-1.236, -0.397) | | |
| $\tau_{int}^2$ | - | 0.198 | - | - |
| | | (0.102, 0.401) | | |
| $\tau_{slo}^2$ | - | 0.309 | - | - |
| | | (0.132, 0.85) | | |
| $\rho_{int}$ | - | 0.48 | - | - |
| | | (0.069, 0.907) | | |
| $\rho_{slo}$ | - | 0.394 | - | - |
| | | (0.022, 0.905) | | |

Table 1 depicts the parameter estimates of all four models with 95% credible intervals. From the credible interval of all the models, it is clear that wind speed does not have any impact on potato production. The rainfall is also insignificant in the independent error model and linear model in the Spatial-temporal framework which matches with the pair-wise scatter plot. However, the ANOVA model and the AR model showed a significance influence of rainfall on response. Raising of temperature will also increase the production of potatoes, on the other hand, the increase in sunshine time will do the opposite.

The MCMC sample that is used to produce model choice criteria is presented in Table 2. Based on the DIC and WAIC, the ANOVA model is the best-fitted model, and based on likelihood value it is said that the Autoregressive model is better. Findings also revealed that the ANOVA model provides the best fit for the sample generated by MCMC. The convergence is

**Table 2. Performance measures of the selected models.**

| Models | Model Choice Criteria | | | | Model Validation Criteria | | | |
|---|---|---|---|---|---|---|---|---|
| | DIC | WAIC | LMPL | loglikelihood | rmse | mae | crps | cvg |
| Independent Error GLM | 4546.793 | 4547.047 | -2273.52 | -2266.42 | 2.060 | 1.707 | 1.154 | 95.37 |
| Linear Model | 4080.063 | 4080.328 | -2040.22 | -2009.82 | 2.038 | 1.681 | 1.03 | 95.37 |
| Autoregressive Model | 4032.988 | 4034.547 | -2017.67 | **-1958.16** | 2.015 | 1.651 | 1.133 | 94.444 |
| ANOVA Model | **4027.623** | **4028.624** | -2014.4 | -1973.67 | **2.013** | **1.644** | **0.932** | **95.64** |

also maximum compared to the other models. So, the subsequent discussion is focused on the ANOVA model. The value of $\rho_S$ is 0.444 which is a good evidence of spatial correlation and is quite higher than the value of $\rho_T$, which indicates that potato production is strongly dependent on spatial variation than temporal effect. The values of $\tau_S^2$ and $\tau_T^2$ is non zero for the ANOVA model from Table 1, which indicates that the spatial and temporal variation exists in the potato production per acre and also establishes the urgency of considering the spatial and temporal domain in the model.

The MCMC iteration produces fitted values for each observed response and the residuals are calculated. These residuals are then aggregated according to the range of the temporal domain to obtain the spatial residuals and standard deviation of the residuals for each spatial domain $d$. Fig 4 illustrates the spatially aggregated residual and SD of the residuals estimated from the spatio-temporal ANOVA model and the plots do not show any significant spatial pattern that needs further investigation. Moreover, Fig 5 depicts the temporally aggregated values for potato production. We used the fitted potato production per acre by dividing the total production by the corresponding area.

The fitted values look quite balanced over the whole time period. No significant overestimate or underestimate is observed from the plot. The lower and upper lines are also quite close to the fitted values, which indicates a good accuracy of the ANOVA model for the Spatial-temporal framework [Fig 5].

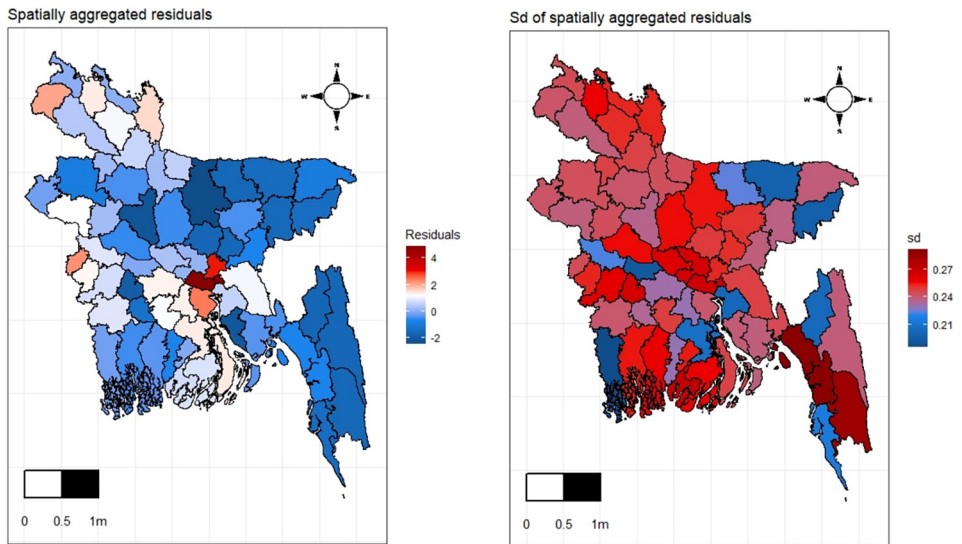

**Fig 4.** District wise residual plot, (a) spatially aggregated residuals, (b) standard error of the residuals.

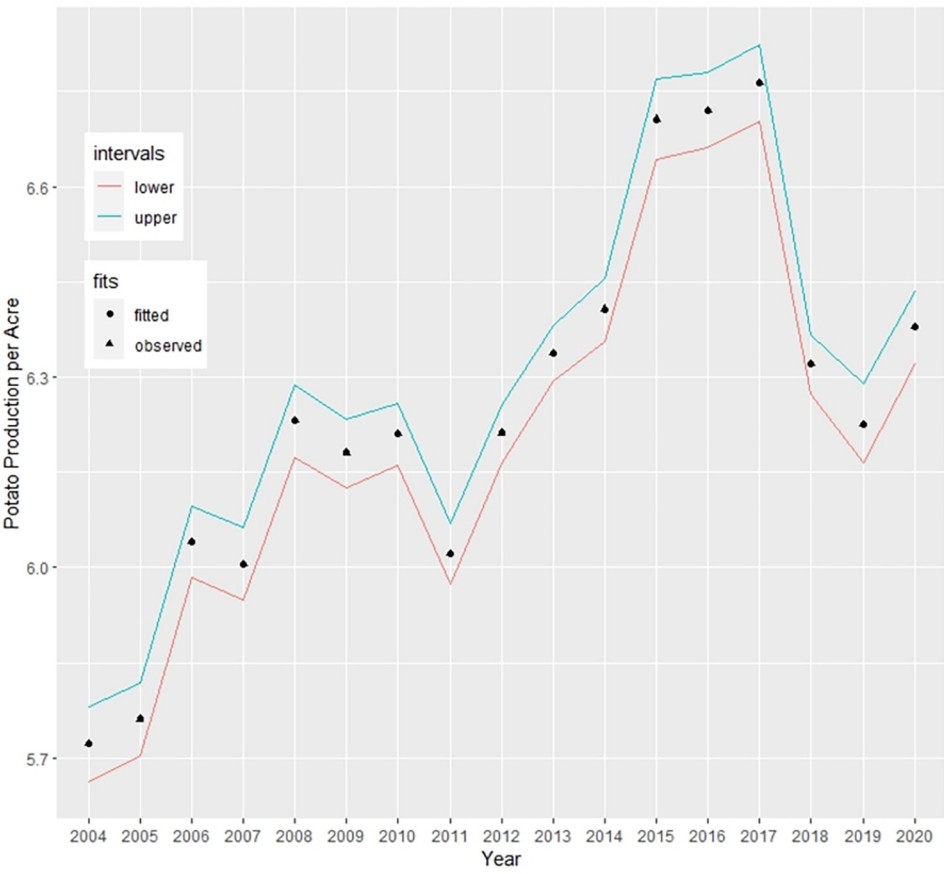

**Fig 5. Time-series plot for observed and fitted potato production with 95% confidence intervals.**

## Discussion

Findings depict that Bangladesh's overall annual potato production has increased over the past 20 years. A previous study predicted that the trend in potato production would be rising [29]. One of the key causes is that the government has been promoting the consumption of potatoes to diversify eating habits and ease the pressure on rice. The potato plant grows in certain weather conditions and this is why the weather indicators have an impact on it in many ways. On the basis of the typical conditions for growing the potato plant, the impact of environmental indicators may be evaluated. The ideal temperature for potato plant development is 20 degrees Celsius [8], while in our region, temperatures throughout the winter months of November through February often hover around 18 degrees. Because of this, the majority of the potato crop in Bangladesh is grown during the winter. In addition, a slight increase in wintertime temperatures will bring potatoes closer to their optimal thermal condition, increasing productivity and correlating with the results of our study. However, according to the Bangladesh Agrometeorological Information portal, both too high and excessively low temperatures harm sprouting as well as the tubers [30]. As the average maximum temperature of Bangladesh in the winter season is below the standard threshold, the model shows the increase of yield as the raise of temperature. Researchers highlighted that dry soil can significantly reduce potato yield or even kill the plant [31]. Although the potato plant needs relatively little water to develop, it does need it for the plant to flourish. Similar inferences can be made from our study

on average rainfall and its mild influence on crop productivity. Only 2 to 4 percent of the yearly rainfall falls during the winter, making it an especially dry season. Researchers pointed out that a dry period and a slight increase in temperature above the margin may have a major influence on the yields of potato tubers [32].

The northeastern region has less sunshine and a lower average temperature than the southwest zone which further illustrates the impact of regional variation in the environment. In addition, there is no production-based geographical fluctuation in the humidity or wind speed. The north and sections of the northeast and northwest of the country are fairly high from the sea level, therefore the weather changes more drastically there than it does elsewhere. Differences in total production are also brought about by these kinds of regional variations in the weather as the temperature is one of the key factors for potatoes tuber which mostly matches with the findings of others [33]. Researchers highlighted that Bangladesh's potato output is influenced by temperature, and the summer months may not be ideal for the irrigation of potatoes due to the high temperatures [34].

Findings revealed that temperature and humidity are significant and it is supported by another study [33]. Insufficient cumulative temperature over the shorter growing periods impacted the growth and yield of potatoes [13]. The ANOVA model and the AR model both demonstrate significant results for the geographical and temporal correlations that have been examined. Although the spatial correlation is larger than the temporal correlation, it is too high for the temporal correlation in the AR model compared with the ANOVA model. This shows that spatiotemporal modeling is valuable for the research. An earlier study noted that the yield of potatoes may be impacted by regional heterogeneity [19]. Commercially viable potato production is influenced by the regional and temporal diversity of soils and agroclimate, as well as the accessibility of water supplies when supplemental irrigation is necessary [35] which quite matches our findings.

The humidity and sunshine is showing a significant negative relationship with the potato yield as the standard relative humidity is below 85% [36] and 6 hours of sunlight. Additionally, Bangladesh's average relative humidity is 78 percent while potatoes are being harvested, meaning that an increase in relative humidity will lower output. In a similar vein, the average amount of sunshine each day in winter is 6.69 hours [37], which is likewise longer than the standard solar time Therefore, productivity was adversely affected by the sun and humidity according to our chosen model.

## Conclusion

The main goal of this research was to determine the factors that affect potato production differently at different periods and regions. Additionally, the authors were keen to investigate how climate variables, such as climate change, affected potato production. The Spatial-temporal model was used to fulfil the objectives of the study in a Bayesian setting. Findings of Bayesian model selection criteria revealed that the Spatial-temporal ANOVA model is more appropriate than other models considered in this study and it has a satisfactory level of convergences on MCMC. Windspeed seems to have no significant influence on potato production, however, temperature, humidity, rainfall, and sunshine have a significant impact on potato production in Bangladesh. The regional variability is more strongly correlated with the production rate than the temporal variation. The authors believed that the findings will be helpful to the policymakers and/or farmers and it is recommended to consider the environmental indicators for potato farming in Bangladesh. Furthermore, the authors are hopeful that the findings will be motivated the researchers who are examining how various climate factors affect different agricultural crop's yields around the world.

## Acknowledgments

The authors are thankful to the academic editor and two reviewers for their valuable comments and suggestions that help to enhance the manuscript's quality.

## Author Contributions

**Conceptualization:** Md. Sifat Ar Salan, Md. Moyazzem Hossain, Imran Hossain Sumon, Md. Mizanur Rahman, Mohammad Alamgir Kabir, Ajit Kumar Majumder.

**Data curation:** Md. Sifat Ar Salan, Md. Mizanur Rahman.

**Formal analysis:** Md. Sifat Ar Salan.

**Methodology:** Md. Sifat Ar Salan, Md. Moyazzem Hossain, Ajit Kumar Majumder.

**Supervision:** Mohammad Alamgir Kabir, Ajit Kumar Majumder.

**Visualization:** Md. Sifat Ar Salan, Md. Moyazzem Hossain.

**Writing – original draft:** Md. Sifat Ar Salan, Md. Moyazzem Hossain, Imran Hossain Sumon, Md. Mizanur Rahman.

**Writing – review & editing:** Md. Moyazzem Hossain, Mohammad Alamgir Kabir, Ajit Kumar Majumder.

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
