## [Decision Letter · Decision Letter 0]

19 Sep 2022

PONE-D-22-21496Measuring the Impact of Climate Change on Potato Production in Bangladesh using Bayesian Hierarchical Spatio-temporal ModelingPLOS ONE

Dear Dr. Hossain,

Thank you for submitting your manuscript to PLOS ONE. After careful consideration, we feel that it has merit but does not fully meet PLOS ONE’s publication criteria as it currently stands. Therefore, we invite you to submit a revised version of the manuscript that addresses the points raised during the review process.

We look forward to receiving your revised manuscript.

Kind regards,

Moumita Gangopadhyay

Academic Editor

PLOS ONE

Journal Requirements:

A clean copy of the edited manuscript (uploaded as the new *manuscript* file).

4. We note that Figures 1, 3 and 4 in your submission contain [map/satellite] images which may be copyrighted. All PLOS content is published under the Creative Commons Attribution License (CC BY 4.0), which means that the manuscript, images, and Supporting Information files will be freely available online, and any third party is permitted to access, download, copy, distribute, and use these materials in any way, even commercially, with proper attribution. For these reasons, we cannot publish previously copyrighted maps or satellite images created using proprietary data, such as Google software (Google Maps, Street View, and Earth). For more information, see our copyright guidelines: http://journals.plos.org/plosone/s/licenses-and-copyright.

a) You may seek permission from the original copyright holder of Figures 1, 3 and 4 to publish the content specifically under the CC BY 4.0 license.  

Natural Earth (public domain): http://www.naturalearthdata.com/.

Additional Editor Comments:

The manuscript entitled ““Measuring the Impact of Climate Change on Potato Production in Bangladesh using Bayesian Hierarchical Spatio-temporal Modelling” (PONE-D-22-21496)” is very important for potato production. Here the authors given a good impression about the current situation of Measuring the Impact of Climate Change on Potato Production in Bangladesh. The findings revealed that there is a potential impact of climate change on potato production in Bangladesh. Bayesian Spatiotemporal modelling for Linear, ANOVA, and Auto-Regressive models was used to find the best fit compared with the independent Error Bayesian model for statistical analysis. However, the authors need to be focus on the following issues for modification-

1. Some specific previous studies on correlation between climate factors and its effect on potato crop (if available) may be included or mentioned in the introduction part.

2. English should be checked by any native speaker.

3. Modification is required for proper synchronization of figures and legends in journal format.

4. Resolution of the images use here should be increased.

5. Needs clearly defined objectives with proper deliverable outcome in details.

Concerning the above-mentioned issues, I am suggesting that this manuscript can be resubmitted as a major revision to this journal.

Reviewers' comments:

Reviewer's Responses to Questions

**Comments to the Author**

1. Is the manuscript technically sound, and do the data support the conclusions?

Reviewer #1: Yes

Reviewer #2: Partly

2. Has the statistical analysis been performed appropriately and rigorously? 

Reviewer #1: I Don't Know

Reviewer #2: Yes

3. Have the authors made all data underlying the findings in their manuscript fully available?

Reviewer #1: Yes

Reviewer #2: Yes

4. Is the manuscript presented in an intelligible fashion and written in standard English?

Reviewer #1: Yes

Reviewer #2: Yes

5. Review Comments to the Author

Reviewer #1: 1. In line no -53 , the font size has to be rectified.

2. Some specific previous studies on correlation between climate factors and its effect on potato crop ( if available ) may be included or mentioned in the introduction part.

Reviewer #2: The author has given a good impression about the current situation of Measuring the Impact of Climate Change on Potato Production in Bangladesh. The findings revealed that there is a potential impact of climate change on potato production in Bangladesh. Bayesian Spatiotemporal modelling for Linear, ANOVA, and Auto-Regressive models was used to find the best fit compared with the independent Error Bayesian model for statistical analysis.

1. English modification has to be done in proper manner.

2. Ligand should be added in figure 1. Addition of standard deviation should added in Fig 1.

3. Explanation of * should be clearly written as figure ligand in Fig 2.

4. Clarity of the Fig 3 should be increased.

5. Discussion part must be more elaborated.

6. PLOS authors have the option to publish the peer review history of their article (what does this mean?). If published, this will include your full peer review and any attached files.

Reviewer #1: No

Reviewer #2: **Yes: **Arpita Das

---

## [Author Response · Author response to Decision Letter 0]

4 Oct 2022

Author responses to the review comments:

We would like to express our sincere gratitude to the two reviewers and the Academic Editor for their valuable comments. We have considered all the comments made by the reviewers and thoroughly revised and formatted the manuscript accordingly. A detailed response to each of the comments is provided in the table below:

Responses to the Academic Editor comments:

Thank you very much. The required files are submitted through the submission system. We include all required information in the cover letter. Revised texts are in red color. 

Responses Journal Requirements:

1. Many thanks. The manuscript is revised according to PLOS ONE’s style. All necessary files are uploaded to the system of the journal. Revised texts are in red color. Page: 1-16

2. Thanks for raising these points. We revised the whole manuscript as per your comments. The whole manuscript is revised in light of grammatical mistakes and typos. A clean version of the manuscript and track changed version is uploaded to the journal system. Revised texts are in red color. Page: 1-16

3. Thanks. We have revised the data availability statement. 

This study is based on the secondary data that is freely available in the websites of the Bangladesh Bureau of Statistics (http://bbs.portal.gov.bd/sites/default/files/files/bbs.portal.gov.bd/page/b343a8b4_956b_45ca_872f_4cf9b2f1a6e0/45%20years%20Major%20Crops.pdf) and Bangladesh Agricultural Research Council (https://bbs.portal.gov.bd/site/page/453af260-6aea-4331-b4a5-7b66fe63ba61/-).

Revised texts are in red color. Page: 16

4. Thank you very much for your concern on this point. We ensure you that all maps are produced by using R-coding written by authors. All of them are the author’s own work. No maps were taken from any other sources.

Responses to the Additional Editor Comments:

1. Thank you very much. We have revised the Introduction section as per comments. Revised texts are in red color. Page: 2-4

2. Thanks. The whole manuscript is revised to fix grammatical issues and typos. Revised texts are in red color. Page: 1-16

3. Thank you very much. All figures and legends are modified as per requirement. All high resolute figures are uploaded to the journal system. Revised texts are in red color. Page: 7-13

4. Thank you very much for highlighted this point. All high resolute figures are uploaded to the journal system separately. 

5. Thanks. We have revised the manuscript and submitted it to the journal system. 

Responses to the Reviewer 1 comments:

1. Thank you very much for your comments and feedback. We revise it. Revised texts are in red color. Page: 3

2. Thanks. We appreciate this comment. We have revised the Introduction section as per your comments. Revised texts are in red color.

Page: 2-4

Responses to the Reviewer 2 comments:

1. Thank you very much for your valuable comment and suggestions that help us improve the manuscript's quality. We have revised the manuscript as per your comments. The whole manuscript is revised to fix grammatical issues and typos. Revised texts are in red color.

Page: 1-16

2. Thanks. Actually, we consider the yearly potato production in Bangladesh. It is a single time series data, therefore, we think that to calculate the standard deviation is not possible here. In the left panel of Fig 1, we visualized the total potato production of Bangladesh by a time series plot and in the right panel, we want to show the spatial distribution of the potato production. Revised texts are in red color. Page: 7

3. Thanks. We add the explanation of * in legend of Fig 2. Revised texts are in red color. Page: 8

4. Thank you very much. We appreciate this comment. All high resolute figures are uploaded to the journal system.

5. Thanks. The Discussion section is revised as per your comment. Revised texts are in red color. Page: 13-16

Finally, the revised manuscript has been produced following the valuable comments and suggestions of the reviewers. Once again, we would like to thank the reviewers for their sincere dedication, professional insights, and earnest cooperation in reviewing the manuscript.

---

## [Editor Report · Decision Letter 1]

1 Nov 2022

PONE-D-22-21496R1Measuring the Impact of Climate Change on Potato Production in Bangladesh using Bayesian Hierarchical Spatial-temporal ModelingPLOS ONE

Dear Dr. Hossain,

Thank you for submitting your manuscript to PLOS ONE. After careful consideration, we feel that it has merit but does not fully meet PLOS ONE’s publication criteria as it currently stands. Therefore, we invite you to submit a revised version of the manuscript that addresses the points raised during the review process.

We look forward to receiving your revised manuscript.

Kind regards,

Moumita Gangopadhyay

Academic Editor

PLOS ONE

Journal Requirements:

Additional Editor Comments:

Dear Sir/Madam

After reevaluating this revised manuscript the following revisions can be taken into consideration-

1. Modifying English Language specially in introduction and conclusion section.

2. Give some justification how this model can be implemented to another agro climatic environment. i.e. Mention global impact of using this model.
---

## [Author Response · Author response to Decision Letter 1]

2 Nov 2022

Dear Moumita Gangopadhyay

Academic Editor

PLOS ONE

We would like to express our sincere gratitude to the two reviewers and the Academic Editor for their valuable comments. We have considered all the comments made by the reviewers and thoroughly revised and formatted the manuscript accordingly. A detailed response to each of the comments is provided below.

Response to the Academic Editor comments:

Thank you very much. The required files are submitted through the submission system. We include all required information in the cover letter. Revised texts are in red color. 

Response to the Journal Requirements:

Many thanks. We checked all the references and ensure that all of them are complete and correct. 

Response to the Additional Editor Comments:

1. Thank you very much. We have revised the Introduction and Conclusion sections as per comments. 

Moreover, the whole manuscript is revised to fix grammatical issues and typos. Revised texts are in red color. 

Page: 1-13

2. Thanks. We add this in the last part of the Introduction section. Revised texts are in red color. 

Page: 4

Thank you very much for your comments and feedback. We already checked all figures using PACE at the time of submission of the revised version of the manuscript (PONE-D-22-21496R1). 

Finally, the revised manuscript has been produced following the valuable comments and suggestions of the reviewers. Once again, we would like to thank the reviewers for their sincere dedication, professional insights, and earnest cooperation in reviewing the manuscript.

---

## [Decision Letter · Decision Letter 2]

8 Nov 2022

Measuring the Impact of Climate Change on Potato Production in Bangladesh using Bayesian Hierarchical Spatial-temporal Modeling

PONE-D-22-21496R2

Dear Dr. Hossain,

We’re pleased to inform you that your manuscript has been judged scientifically suitable for publication and will be formally accepted for publication once it meets all outstanding technical requirements.

Kind regards,

Sathishkumar V E

Academic Editor

PLOS ONE

Additional Editor Comments (optional):

Reviewers' comments:

Reviewer's Responses to Questions

**Comments to the Author**

1. If the authors have adequately addressed your comments raised in a previous round of review and you feel that this manuscript is now acceptable for publication, you may indicate that here to bypass the “Comments to the Author” section, enter your conflict of interest statement in the “Confidential to Editor” section, and submit your "Accept" recommendation.

Reviewer #3: All comments have been addressed

Reviewer #4: (No Response)

2. Is the manuscript technically sound, and do the data support the conclusions?

Reviewer #3: Yes

Reviewer #4: (No Response)

3. Has the statistical analysis been performed appropriately and rigorously? 

Reviewer #3: Yes

Reviewer #4: (No Response)

4. Have the authors made all data underlying the findings in their manuscript fully available?

Reviewer #3: Yes

Reviewer #4: (No Response)

5. Is the manuscript presented in an intelligible fashion and written in standard English?

Reviewer #3: Yes

Reviewer #4: (No Response)

6. Review Comments to the Author

Reviewer #3: All the comments as mentioned in the last review, now addressed and now the paper stands Accepted with no further revisions.

Reviewer #4: (No Response)

7. PLOS authors have the option to publish the peer review history of their article (what does this mean?). If published, this will include your full peer review and any attached files.

Reviewer #3: No

Reviewer #4: **Yes: **Usha Moorthy

---

## [Editor Report · Acceptance letter]

11 Nov 2022

PONE-D-22-21496R2 

Measuring the Impact of Climate Change on Potato Production in Bangladesh using Bayesian Hierarchical Spatial-temporal Modeling 

Dear Dr. Hossain:

I'm pleased to inform you that your manuscript has been deemed suitable for publication in PLOS ONE. Congratulations! Your manuscript is now with our production department. 

Kind regards, 

on behalf of

Dr. Sathishkumar V E 

Academic Editor

PLOS ONE